# Parental Beliefs and Feelings about Autism Spectrum Disorder in Iran

**DOI:** 10.3390/ijerph17030828

**Published:** 2020-01-29

**Authors:** Sayyed Ali Samadi

**Affiliations:** Institute of Nursing and Health Research, Ulster University, BT521SA Coleraine, Ireland; s.samadi@ulster.ac.uk

**Keywords:** autism spectrum disorders, parental beliefs, parental feelings, Iran

## Abstract

Background: This study provides information on beliefs that parents of children with autism hold in Iran. The main focus is on their beliefs about the cause and the way that this condition is explained based on the first signs that made them be concerned for their children. Method: To attain the aims of this study, 43 parents of children with ASD (27 mothers and 16 fathers) were recruited and interviewed in two sessions in their home. A mixed method approach was used to understand Iranian parents’ reaction to receive diagnosis for their children. Results: Based on findings for the identification, description, and treatment of ASD in Iran, it is argued that since Iranian parents had their special justification regarding their experience with ASD, early child development and interventions must be understood within cultural context. Culturally informed research on ASD is vital to boost awareness of the importance of understanding parental concerns and their need for educational and psychological services in countries in which autism is less known, undiagnosed, misdiagnosed, or even stigmatized. Understanding the difference in ASD meaning across cultures urges stakeholders such as service providers and policymakers to accept and appreciate cultural and individual diversities in the present century.

## 1. Introduction

Autism Spectrum Disorder (ASD) represents a group of neurodevelopmental disorders marked by impairments in communication/social interactions and special patterns of behaviours with onset during early stages of development [1]. Although causes of ASD are not fully understood [2,3] it has been the subject of intense inquiries [4,5]. The genetic factors are an important possible source of causation [6]. However, ASD might be prone to parents’ ideas and beliefs on its aetiology and causes [7] because there is an absence of a justifying theory on the neurobiological mechanisms of this diagnosis [8].

Although many researchers speculate that the incidence of ASD is constant across cultures, there is a dearth of studies in many western [9] and nonwestern countries [10]. Most of the present literature on ASD is done in western countries [11,12]. Although professionals have stressed the need for an increase in awareness of cultural influences on ASD [13,14], little is known about ASD within a cultural context and the way it is understood in other cultures [15]. Kim [16] in a comparative study using field notes and semi-structured interviews with families of children with ASD in three different countries consisting of Canada, Nicaragua, and Korea found that ASD is socially and differently constructed in different cultures. 

The reaction families show about ASD diagnosis and treatment are possibly influenced by their cultural background. Based on Helms and Cook, [17] culture can be defined as ‘‘the values, beliefs, language, rituals, traditions, and other behaviours that are passed from one generation to another within any social group.” Culture is also defined as a dynamic, yet stable set of goals, beliefs, and attitudes shared by a group of people [18]. This suggests the need for cross-cultural studies to test the generality of the current concept of ASD [19] because as the present wealth of data suggest, culture shapes a family’s particular beliefs about ASD [20]. One way to investigate parents’ beliefs on ASD is to explore their explanations of the situation that they are living in. 

Parental beliefs on ASD have been shown to impact their intervention choices [21,22] on the other hand, one way to investigate parents’ reviews of their situation as caregivers of a child with ASD is to explore their explanations and understanding of this diagnosis. Specifically, parents’ beliefs about ASD shape their explanation of signs manifestation, the time that they take to seek out intervention, and the type of intervention they decide to have for their children [23]. Nonetheless, parental perspective and beliefs system about the early signs of ASD and paying attention to them allows the possibility of targeted intervention or more careful monitoring [24]. As an example, families who believe that their children’s autism is due to a gastrointestinal problem are more prone to choose a diet-focused treatment approach rather than a behavioural approach [23]. Given the variability in the interpretation of ASD across cultures, it is important for service and support providers, practitioners, and policymakers to develop cross-cultural competence and sensitivity to serve children with ASD and their families.

Based on findings by Bagatell [25], in the western culture, ASD notions are largely under the influence of the biomedical community and is considered as a deficit and deficits need to be “fixed.” Whereas, in Indian culture, *Karma* and fate/destiny were reported to be the dominant notion in the community [26,27] using treatment from traditional Indian culture. While in a more recent study about Indian parents’ beliefs who were living in the Western culture, it reported that the traditional belief was absent and as a consequence, they generally used a combination of behavioural and biomedical therapies for their children, including some Western complementary and alternative treatments [23]. Although there is a dearth of studies in this field, Gray’s study [7] on lay conceptions of ASD among Australian parents showed that they expressed verities of explanations to account for their children’s diagnosis, such as birth trauma and illness during pregnancy. Some of the cited beliefs related to magical or religious beliefs and in their description obvious features were indicating a sense of guilt. Some parents even thought that ASD might be the result of a past mistake or karma, a similar finding reported with Indonesian parents [28]. Mercer et al. [29] reported that even parents who attribute their child’s ASD diagnosis to a biomedical factor expressed a sense of guilt for being responsible to passing on genes that caused this diagnosis for their children. These different beliefs and understanding of ASD were rooted in the parental cultural background and unique experiences as a caregiver of a child with ASD.

The present study investigates Iranian parents of children with ASD, beliefs and ideas about the causes of this diagnosis that are directly rooted from their experience with ASD, as well as the way that the condition and its prognosis are understood by them.

Findings by previous studies showed a link between parental beliefs and their ideas on the possible aetiology which most of the time impacts on the treatment choice for children and the child’s integration into the surrounding community [9,30,31]. The changing of perspective to embrace children with ASD in a full manner is considered to be the responsibility of societies from a group of parents’ perspective [32]. These findings stress the impact of cultural clashes for the service providers to help parents receive suitable therapy to continue their growth and development [33]. It also helps the profession become more culturally aware and to focus on diversity both within and across cultures regarding ASD.

To understand the complex interplay between contributing factors of the challenges associated with the parental understanding of ASD, the present study considered the ecological model [34] in which the main endeavour is on considering the impacts of the multiple ecological environments on each other. Therefore, according to an ecological model, the focus is on a transformed ecology in which children with different types of disabilities can develop through the interaction of their skills with a responsive context at different levels [35]. 

## 2. Materials and Methods 

### 2.1. Procedures

#### 2.1.1. Sample

A group of Iranian parents of children with ASD were recruited from different services to understand their experience and the way that they explain their impression about this diagnosis for their children. The recruitment was done in Tehran the capital city of Iran. Each family had two visits. They were interviewed about their ideas on the causes of ASD, the first signs that made them concerned about their child, the reason for their concern, the name that they think describes their child’s condition properly, and the age that they received the diagnosis for their children. Structured interviews were conducted with parents in their home.

Each family had two sessions and the duration of the sessions was around 2 h. In the first session parents were provided with the information sheet and data collecting questionnaires and in the second session the interview was done and the previously provided questionnaires were collected. Therefore, the first session was devoted to informing parents about the study, its aims, answering their possible questions, and giving them the demographic questionnaires about themselves and their children and helping the parents get acquainted with the researcher and the study.

The qualitative interview was carried out during the second session when parents were already informed about the study through the explanation which was offered through the information sheet and word of mouth in the first session. The interview time was around 30 to 60 min. 

A group of 43 volunteer parents from different services in Tehran were contacted. They consisted of 16 (37.2%) fathers and 27 (62.8%) mothers. They were caring for their child with ASD in ages ranging from 3 to 17 years with a mean of 8.2 years. Recruitment was made by identifying those parents who had a child with confirmed ASD from Iranian Special Education Organisation (ISEO) or other related official organisation such as Iranian Social Welfare Organization (ISWO) with a confirmed diagnosis of ASD and a registered profile. This information was sourced from the Special schools for ASD children in Tehran and through the Mother and Child clinics in different parts of Tehran. The head of these centres was contacted in person to explain the study and to get permission to participate in the weekly or biweekly parental sessions at the centre and to distribute announcements to recruit volunteer parents. Participants in this study had to meet the following criteria:

Caregiving for children with a confirmed and registered diagnosis of ASD as the main diagnosis;

Caregiving for children with ASD aged between 3 and 17 years

living in Tehran city; 

Caregiving the child with ASD at the time of the study.

#### 2.1.2. Ethical Issues

Ethical approval was sought from the Ulster University and the reference number 07/0143 was allocated as approval to this project. Consent forms were signed in the first session. Parents were informed that they have the right to withdrawal at any time during the study. 

### 2.2. Study Design

A mixed-method approach was considered for this study in which a semi-structured interview schedule was prepared for the qualitative part, and parent and child demographics questionnaires were developed for the quantitative part. 

### 2.3. Instrument

To collect the data for the qualitative part an interview schedule was developed. The interview consisted of six open-ended questions. The questions were about parents’ idea on the problem with their child, the first signs which made them concerned and the child’s age when the sign emerged, parental expectations following the diagnosis, and the way that they define Autism and its causes. To be able to understand the parental idea on the questions general probes were used (How would you describe/explain the problem? Could you give me more detail about that? Could you give an example?). Each interview was proceeding similarly for each family.

To collect data for the quantitative part two questionnaires were developed. The first questionnaire consisted of 20 questions on parental demographics and the second one consisted of 10 items.

### 2.4. Data Analysis

In complex studies that deal with the reality of peoples’ lives, their health, and social environments, combining research methods may assist researchers to seek ‘more complete’ understanding [36]. Parents’ ideas on ASD have different dimensions and could be classified as a complex field of study, hence combining both a qualitative and quantitative approach in the study may be of particular use when investigating the complexity of such issues. 

A thematic analysis approach was adopted and the interviews were transcribed verbatim and were categorized according to their thematic contents. An Independent rater re-categorized the parental responses and 95% of consensus between two independents were reported. Counting and using percentage and frequencies was considered to be integral to the analysis process in qualitative studies and numbers was used to establish the significance of findings even with the qualitative method of analysis [37].

To evaluate the statistical significance relationship between investigation factors since the data were at ordinal and nominal levels nonparametric statistics were used.

## 3. Results

### 3.1. Quantitative Finding

Demographic data on parents and children are presented in Table 1 and Table 2.

To understand the possible relationship between collected demographic data and the categorical factors several comparisons were performed.

### 3.2. Child’s Age When Parents Became Concerned 

Regarding the age that parents noticed the sign, 14 (32.6%) parents said that they became worried about their child before he/she was one year old; nine (20.9%) said that they identified a problem before the second year of age, and a further 18 (41.9%) parents became worried by the sign they saw before the third year of their child’s age. Only two (4.7%) parents identified the signs after the third year of their child’s life. No relationship was found between the mothers and fathers (χ2 = 2.89; df = 1; *p* = 0.089), their age and age of identification of the child’s problem (χ2 = 0.7; df = 1; *p* = 0.78) (χ2 = 0.15; df = 1; *p* = 0.69), or mothers’ (χ2 = 0.56; df = 1; *p* = 0.45) and fathers’ education (χ2 = 0.004; df = 1; *p* = 0.94), mothers’ employment status (χ2 = 0.4; df = 1; *p* = 0.52), having a relative living with the family (χ2 = 0.93; df = 1; *p* = 0.33), and relative living nearby the family (χ2 = 2.16; df = 1; *p* = 0.14).

### 3.3. The Age That the Child Received the Diagnosis of ASD

Parents were asked about the age of their child when they *received* the diagnosis of ASD. Three children (7%) received the diagnosis when they were one year old, four (9.3%) when they were two years, 19 (44.2%) when they were three years, and 12 (27.9%) when they were four years, and finally four (9.7%) when they were five. No significant relationship was seen between the child’s age when the first signs were seen by parents and when they received the diagnosis (χ2 = 0.06; df = 1; *p* = 0.8), between the mothers’ and fathers’ age and age of identification of the child (χ2 = 0.38; df = 1; *p* = 0.53) (χ2 = 1.34; df = 1; *p* = 0.24), or mothers’ (χ2 = 0.29; df = 1; *p* = 0.59) and fathers’ education (χ2 = 0.16; df = 1; *p* = 0.23). No significant relationship with the mothers’ employment status (χ2 = 0.005; df = 1; *p* = 94), having a relative living with the family (χ2 = 6.67; df = 1; *p* = 0.01) and relatives living nearby the family (χ2 = 0.75; df = 1; *p* = 0.38).

No statistically significant difference was seen between different factors such as parents’ gender (χ2 = 2.89; df = 1; *p* = 0.08), studies (Mothers: χ2 = 2.08; df = 1; *p* = 0.149; Fathers: χ2 = 0.420; df = 1; *p* = 0.51), or socioeconomic level based on the area that parents live in (χ2 = 4.85; df = 1; *p* = 0.18), and ideas about ASD.

### 3.4. Qualitative Finding

#### 3.4.1. Presence of the Problem

Parents were asked if they think that there is a problem with their child. They all (n = 43, 100%) confirmed their child’s problem. Twenty two (51.6%) used phrases such as “of course, one hundred percent, or yes he/she has (many) problem(s)”, and “yes this is why I accepted to be interviewed”, 20 (46.5%) confirmed using “yes” without any comments and only one (2.32%) mother was not so sure and said “*We think that he has problem*”. 

#### 3.4.2. The Way That Parents Describe Their Child’s Present Problem

The parental explanation of the problem was also questioned. Table 3 depicted a categorised list of parental explanation about the problem.

More than one-third of the parents (n = 18, 41.8%) believed that their child’s problem was limited ability in social/communication and understanding of the social context and demands. The second most mentioned items were behavioural problems, improper social behaviours which draws other people’s attention (strange head and arms movement, difficulty in looking at the other people’s eyes and face). Some parental quotes were as follows (“M” stands for mothers’ quotes and “F” for fathers).

[M.21] "His problem is that he is “present” and at the same time does not! It is not like the presence of an ordinary child by his mother. He has no communication and relationship with others. I do not feel the real sense of motherhood and there is not a real mother and child relationship between us”.

[F.39] "To me, it is just like a computer which is infected by a virus. We need to discover this virus and find an untie virus for it. This make the computer work properly".

#### 3.4.3. The Name that Parents Think Describes the Problem

Parental ideas on the name that describes the situation of the children properly, showed that 18 (41.8%) said that “Autism” and “Asperger’s” best describe the problem. Eight parents (18.6%) instead of naming the problem, gave an example. One mother said that “*My daughter has a cloudy brain something like a fog which made understanding the world difficult for her*". Another mother said, "*Someone kept inside her/himself, isolated and solitaire*”. A mother said that “*I call it a disaster! The most difficult problem and tragedy that ever could happen! These are the names I used to address it*”. Another mother said that “*I think "Autism" or whatever they use to explain this problem is painful. Especially because he is growing up and this make things more difficult*”. A father said that “*This is a dilemma or strange puzzle, something you can never solve like a closed maze with no way to exit*”. Seven (16.2%) parents used “Mentally disabled or disturbed”. Seven (16.2%) others said that they use no name, and two (4.96%) parents said that “Behavioural problem” better explain their child’s problem, whereas one (2.32%) used a “Genetic deficit” to name this diagnosis. 

#### 3.4.4. First Signs Which Made Parents Concerned About Their Child

Parents were asked about the first signs which caused them to bring their child for an assessment. Table 4 categorized their answers. 

Parents in each category gave some examples. Sixteen (37%) parents named one indicator, 18 (41%) parents mentioned two, six (13.9%) mentioned three, and three (9.3%) parents named four factors. 

The most frequently mentioned problem was difficulties with language and speech. The next most mentioned item was behavioural problems which was explained in examples such as restlessness, hyperactivity, head bouncing, and continuous crying. 

### 3.5. Causes of ASD

Table 5 has categorized parents’ ideas about the causes of ASD.

Parents in each category, gave some example to explain their answers. Thirty-nine (90.6%) considered only one factor as the cause of ASD, whereas three (6.97%) parents mentioned two factors, and only one (2.32%) parent nominated three factors as the possible causes of ASD.

[F.10] “A physical base and a problem in his brain or because of air pollution or chemical effects on his mind. We were attacked by chemical weapons while I was in the front line during the Iran and Iraq war. I must say that we have two healthy children and they are so bright and intelligent”. 

[F.17] “Maternal stress and anxieties cause the problem. My son’s problem could be due to our family’s lack of normal communication with others and social limitations imposed on us and also “chlorosis” or other infections”.

[M.31] “I think generally some environmental factors cause this problem. In my son’s case, it was because of loud noise, a very loud noise from our TV when he was so close to it. He was shocked and paralysed because of it”.

[M.32] “It is just because of people’s sins and underrating the moral codes, we may have done something wrong unconsciously. I am sure it is the reason but I do not know what the wrong action was it. I and my husband are so faithful to the religious rules and we are trying to consider and follow them all”. 

[F.39] “It is because of parents’ sins. You see, children get their appearance form the mother and the spirit from the father, my son perception and cognition defected so I must be blamed because of it. This is my fault, not my wife’s!” 

### 3.6. Change in Parental Expectations After the Diagnosis

When parents were asked if they changed their expectations after their child’s diagnosis, 30 (69.7%) said that they reduced the level of their demands. 

[M.31] “How could we stop changing our expectations for him? It was so disappointing for me, our son was about to inherit the family value and name, we are a minority (Zoroastrian) and he is the only son and should carry the family name and give it to his son, but none of them could ever happen and everything vanished for our family which made us very sad”.

[M.43] “We changed our expectation for her drastically. She is a disabled girl in our society; nobody knows what will happen to her when I and her father passed away. We have to do all our best to cure her and help her to behave like other girls at her age”. 

Eight (18.6%) said that they did not change their expectations. 

[F.17] “I did not change my expectations because I knew that they were wrong and my son’s problem was not very serious”. 

Four (9.30%) parents said that they let their child do what they think he/she can do and help in what they think could be challenging.

[M.30] “At the beginning, I changed my expectations and I had no hopes. I thought that he is not able to do anything and I ignored my desire, but then I became more aware of his potential and helped him to gain abilities”.

One (2.32%) parent said that he has increased his expectations because of the increased levels of efforts imposed on him compared to the other parents.

[F.26] “We faced a very high wall in front of us, but we did not surrender, we have done our best so I have increased my level of expectations because we are doing more than other parents for their normal children, so we want more”. 

## 4. Discussion

There are many undiscovered aspects of ASD. Parental experience and their ideas and explanation which are rooted in their culture and communities are among those unknown aspects which surrounded ASD and deserve to be understood. Available data indicates parents in different cultures pose different ideas about ASD [38]. To be able to provide satisfactory support and services for children with ASD and their families, professional and general health service providers must acknowledge how culture can impact perspectives about ASD [20]. The benefit of this understanding in addition to considering communities’ diversity aids professionals working with families of children with ASD as Kluth [39] suggested to recognize and preserve the families’ dignity, accommodate services and resources properly, set reasonable and progressive goals, value and recognize the wisdom of the family. Although parents in this study were from one major city in Iran, they echoed diverse attitudes based on their experience, culture, and the communities they belong to. This highlights the need for professionals in health care, education, and related support and services to be aware of this diversity among the families of people with ASD and try to achieve culturally competent and sensitive practices. This is not an easy task because as Ravindran and Myers [23] reported in their study with a group of Indian families of children with ASD living in the western culture, parents from different cultural groups might keep their traditional beliefs to themselves when interacting with professionals in a dominant culture.

Iranian parents in this study generally became concerned about their children’s development at an early stage of development. More than half of the parents in this study identified their child’s problem before their second year of life. This is similar to what is reported in other countries [40]. 

Parents in this study described their children’s problem as a social communication problem, this finding is in congruence with reports from South-Eastern Asian Muslim families that immigrated to the United States [15]. Parents in this study generally used the term ‘autism’ as a name to describe their child’s condition. Moreover, it was revealed that the absence of appropriate communication with others is an important feature of ASD for the Iranian parent. This, in turn, could cause a delay in diagnosis of ASD for the child because lack of communication with others, instead of being recognised as a sign of a developmental problem, is often attributed to a personality trait in the child or his/her gender [41]. Communication deficits and behavioural problems cannot be fully observed until the child has the opportunity to communicate with others in different situations or environments. Findings show that uneven skill development of children with ASD and the presence of special behavioural patterns only becomes apparent when the child is put into a new environment that may hinder the diagnosis of ASD in the child’s early years.

Regarding the causes and aetiology of ASD in their offspring, Iranian parents mentioned different factors, but maternal factors during the pregnancy such as maternal stress, fears, delivery problems, and craving were dominant ideas (N = 16, 37%). 

Iranian parents in this study generally attributed the cause of ASD in their child to mothers and their conditions. There were also five Iranian parents (12%) who mentioned spiritual or religious factors as the main cause of ASD in their child. This was opposite to what Mercer et al. [29] reported. According to their study with 41 Canadian parents, most of them (N = 37, 90%) believed that genetic reasons contributed to ASD in their children, whereas only three (7%) of the Iranian parents mentioned a genetic link as the cause of ASD. However, Harrington et al. [31] reported that in a sample consisting of 71 American parents of children with ASD, 16 (26%) considered there was a genetic predisposition to ASD and another group (N = 18, 29%) believed that immunizations were the cause of ASD. Studies of parental beliefs about the causes of disabilities in some other non-Western societies showed similar findings to Iran. Park et al. [42] found that in Vietnam the dominant view about the causes of disability is an ancient belief that disability is attributed to sins or immoral deeds committed by the afflicted person’s family or even ancestors. This view, then, involves cultural shame, blaming family members, and individuals with disabilities. This could be considered as a reason for Iranians to view disabilities as a stigma that brings shame and blame for parents and siblings, as reported in some other studies [43]. 

Addressing the cultural knowledge and competencies of ASD service providers in both pre-service and in-service field of professional development might be considered as the main implication of the present findings.

In the present study, only 5% of Iranian parents became concerned about their children while they were three years and older and the rest became concerned before this age. This is congruent with international reports [44]. 

No statistically significant difference between the age of the child when parents became concerned about them and the factors such as parental gender, their education, and age was discovered. A similar insignificance was reported in western reports [45].

Almost 80% of the children received a diagnosis for ASD after they were three years old and there is an obvious gap between the age of children when their parents became concerned about them and the age when they received a diagnosis and it was congruent with reports from other countries [44] and factors such as parental education and age. Finally, there was no statistical significant difference between factors such as parents’ gender, their education, socioeconomic level, and ideas about ASD which indicate the dominance of cultural beliefs and as Jegatheesan et al. [33] indicated acknowledgement or understanding of the parental cultural conception of developmental disability, as a sacred obligation, could have avoided misunderstandings around the aims of assessment and choice for intervention. Therefore, at the very basic level, findings of the present study might boost understanding of parental beliefs and ideas on ASD at a multicultural level and help the different ASD services providers predict justification and reasoning from a different lens.

## 5. Limitations

The present finding was obtained from parental interviews and quantitative data obtained from a relatively small group of parents. These volunteer parents were predominantly from a middle and upper-income social class who were urban residents and attending mother and child clinics and parental biweekly educational sessions in special schools and clinics for their children. These findings, therefore, echo resourceful and actively engaged parents and do not necessarily reflect the ideas of other Iranian illiterate or less educated, rural resident groups. Since ideas and impacts on ASD may differ in other socioeconomic classes, future studies should target parents of children with ASD in other populations using bigger samples and other data collecting instruments that measure or address parental beliefs about autism to understand differences and similarities, and what may be the point if similar ASD has a communication and behavioural aspect which are most likely nonverbal across all cultures.

## 6. Conclusions 

This study explores the impact of Iranian parents of children with ASD belief and perspective based on their experience and the way that they are defining this diagnosis. The implication of the finding could help professionals such as mental and general health service providers to effectively take into account different aspects of social and environmental milieus such as culture and their communities when dealing with parents of children with ASD and the children themselves. It seems that cultural groups concerning ethnical and racial entity in the present literature of ASD have a very broad and general description [20]. Considering cultural groups, with a focus on ethnicities and racial groups might be able to balance this dominance. Since the causal origin of ASD is unclear, therefore, many diverse cultural groups have their justifications about the way that the symptoms emerge.

Observing and experiencing the difference in ASD meaning in different cultures urges professionals and policymakers to accept and appreciate cultural diversity and even individual differences or ‘differ-perspective and understanding’ in the present century.

Education at different levels for parents and providing information on ASD might be a major contributing factor towards the reduction of the stigma attached to the disability and difference in general.

## Figures and Tables

**Table 1 ijerph-17-00828-t001:** Parental demographic data.

Variable	Frequency	Percentage
Parents completing scales	Mothers (27)	62.8%
	Fathers (16)	37.2%
Mothers’ age range	Under 30 (3)	7%
	30–39 (27)	62.8%
	40–49 (12)	27.9%
	50–59 (1)	2.3%
Fathers’ age range	Under 30 (3)	7%
	30–39 (16)	37.2%
	40–49 (16)	37.2%
	50–59 (8)	18.6%
Mothers’ education	2 (Middle school graduate)	4.7%
	24 (High school graduate)	55.8%
	17 (University graduate)	39.5%
Fathers’ education	3 (Middle school graduate)	7%
	17 (High school graduate)	39.5%
	23 (University graduate)	53.5%
Mothers’ ethnicity	Fars (31)	72.1%
	Azeri Turkish (9)	20.9%
	Armenian (1)	2.3%
	Other Iranian (2)	4.7%
Fathers’ ethnicity	Fars (31)	72.1%
	Azeri Turkish (6)	14%
	Kurdish (3)	7%
	Armenian (1)	2.3%
	Other Iranian (1)	2.3%
	Non-Iranian (1)	2.3%
Single parents	Yes (3)	7%
	No (40)	93%
The parent whom the child lives with	Mother (4)	9.3%
	Father (0)	0
	Both (39)	90.7%
Number of family members	Less than 3 (1)	2.3%
	3 (17)	39.5%
	4–5 (18)	41.9%
	More than 5 (7)	16.3%
Parents’ family relationship (marriage between family members)	Yes (8)	18.6%
	No (35)	81.4%
Child primary carrier	Mother (34)	2.3%
	Father (1)	79.1%
	Both (8)	18.6%
Main wage earner of the family	Mother (1)	2.3%
	Father (31)	72.1%
	Both (11)	25.6%
Parents’ living places	Private (26)	60.5%
	Rental (11)	25.6%
	With parents (4)	9.3%
	Government houses (2)	4.7%
Years parents stayed in their present address	Less than 2 years (9)	20.9%
	Between 2 to 5 years (11)	25.6%
	Between 5 to 10 years (16)	37.2%
	More than 10 years (7)	16.3%
The district which parents were living in Tehran	Centre (11)	25.6%
	North (11)	25.6%
	South (5)	11.6%
	East (5)	11.6%
	West (10)	23.3%
	Out of Tehran (1)	2.3%
Dependents living at home with the family (grandparents, aunt, uncle)	Yes (12)	27.9%
	No (31)	72.1%
Relative lives near the family	Yes (25)	58.1%
	No (18)	41.9%

**Table 2 ijerph-17-00828-t002:** The children demographic data.

Variable	Frequency	Percentage
Children’s age	3–7 (19)	44.1%
	8–17 (24)	55.8%
Children’s gender	Girls (11)	29.7%
	Boys (26)	70.3%
Birth order	Fist Child (24)	55.8%
	Second Child (12)	27.9%
	Third Child (6)	14%
	Forth Child (1)	2.3%
Type of the schooling which children attend	Special School (19)	44.2%
	Special Unit (Clinic) (19)	44.2%
	Mainstream School (5)	11.6%
Homeschooling	Yes (17)	39.5%
	No (26)	60.5%
Verbal communication of children	Yes (26)	60.5%
	No (17)	39.5%

**Table 3 ijerph-17-00828-t003:** The present problem of the children from the parents’ perspective.

Problem	*n*	Percentage
Communication and social relationship	18	(41.8%)
Behavioural problems	9	(20.9%)
Language and speech	7	(16.2%)
Learning disabilities and mental problems, inability to understand	6	(13.9%)
Giving an example to elicit their view	3	(6.97%)

**Table 4 ijerph-17-00828-t004:** The child problem indicators from the parental perspective.

Indicators	*n*	Percentage
Speech delay	13	(30.2%)
Strange behaviour, head bouncing, continuous crying, restlessness	9	(20.9%)
Motor problems delay	5	(11.6%)
Hearing problem	5	(11.6%)
Communication problems, eye contact	4	(9.30%)
Seizures	4	(9.30%)
Sudden regression	3	(6.97%)

**Table 5 ijerph-17-00828-t005:** Parents’ ideas about the causes of ASD.

Cause	*n*	Percentage
Maternal situation (stress, fears, delivers problems, craving, etc.)	13	(30.2%)
Having no information (having no idea, do not care)	9	(20.9%)
Brain or body-based problem	7	(16.27%)
Environmental factors (pollution, chemicals, electromagnetic waves, etc.)	6	(13.9%)
Results of spiritual and religious factors	5	(11.6%)
Combination of different factors	1	(2.32%)
The special social situation of the family (having no relationship with the others)	1	(2.32%)
Heredity (genetics)	1	(2.32%)

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
