# Peer review of "Parental Beliefs and Feelings about Autism Spectrum Disorder in Iran"

_ijerph, 2020, doi:10.3390/ijerph17030828_

Round 1

Reviewer 1 Report

The manuscript “Parental beliefs and feelings about Autism Spectrum Disorder in Iran” focus on an interesting topic both from a clinical and a research perspective. Although the study is relevant and has several strengths, different suggestions and recommendations have included in the following lines.

Please revise the writing and punctuation marks through the whole text, there are some commas missed along the text and mistake. For instance and among others: Table 4 should include capital letter in the first letter of the title.

Although some information is included, there is a lack of theoretical information, in my opinion, about the consequences of the parents’ ideas about ASD. More studies could be included in this part about what are the implications of the parents’ beliefs about ASD.

In the Method section, I would include the recruitment part as sample or participants section (APA). The sample part should be included in the procedure, as authors are explaining how the conducted the research.

Also, it could be interesting to add a specific part for instruments and another one for data analysis.

With regards to data analysis, authors mention that they used a mixed-methods approach. However, authors only explain the qualitative part. No information can be seen about the qualitative part. From my point it could be relevant to include some analysis that allow seeing the relation between parents’ gender, studies or socioeconomic level and ideas about ASD.

Finally, the discussion could benefit of the mentioned in the previous point about more studies regarding consequences of parents’ believes as well as the inclusion of qualitative methods. Otherwise limitations should be included about the lack of a quantitative approach   

Author Response

The manuscript “Parental beliefs and feelings about Autism Spectrum Disorder in Iran” focus on an interesting topic both from a clinical and a research perspective. Although the study is relevant and has several strengths, different suggestions and recommendations have included in the following lines.

R: Thank you for the positive starting point and the following useful comments

Please revise the writing and punctuation marks through the whole text, there are some commas missed along the text and mistake. For instance and among others: Table 4 should include capital letter in the first letter of the title.

R: The Manuscript was revised from the typos and punctuation aspect and corrections were made 

Although some information is included, there is a lack of theoretical information,

R: A paragraph one theoretical framework and one other paragraph on the consequence of parents’ ideas on ASD along two other references added:

The present study considered theoretical framework in attempting to understand the complex interplay between contributing factors of the challenges associated with parental understanding of ASD.

The ecological model (Bronfenbrenner, 1979) main endeavour is on considering the impacts of the multiple ecological environments. Therefore, according to an ecological model, the focus is on a transformed ecology in which children with different types of disabilities can develop through interaction of their skills with a responsive context (Turnbull et al. 1999).

Bronfenbrenner, U. (1979). Contexts of child rearing: Problems and prospects. American psychologist, 34(10), 844.

Turnbull, A. P., Blue-Banning, M., Turbiville, V. & J. Park (1999). From Parent Education to Partnership Education: A Call for a Transformed Focus. Topics in Early Childhood Special Education, 19(3):164-171.

in my opinion, about the consequences of the parents’ ideas about ASD. More studies could be included in this part about what are the implications of the parents’ beliefs about ASD.

R: Other studies on the implications of the parental beliefs were added (three references added)

Harrington, J. W., Patrick, P. A., Edwards, K. S., & Brand, D. A. (2006). Parental beliefs about autism: Implications for the treating physician. Autism, 10(5), 452-462.

Jegatheesan, B., Miller, P. J., & Fowler, S. A. (2010). Autism from a religious perspective: A study of parental beliefs in South Asian Muslim immigrant families. Focus on Autism and Other Developmental Disabilities, 25(2), 98-109.

Johnston, C., Seipp, C., Hommersen, P., Hoza, B., & Fine, S. (2005). Treatment choices and experiences in attention deficit and hyperactivity disorder: Relations to parents’ beliefs and attributions. Child: Care, Health and Development, 31, 669–677.

In the Method section, I would include the recruitment part as sample or participants section (APA). The sample part should be included in the procedure, as authors are explaining how the conducted the research.

R: The proposed amendments were considered in both recruitment and sample part and Recruitment as a separate subtitle deleted and merged with sample based on the recommendation and all this sub-titles were put under the Procedure under the Method section.

Also, it could be interesting to add a specific part for instruments and another one for data analysis.

R: Two different parts under the Instruments and Data Analysis were added:

With regards to data analysis, authors mention that they used a mixed-methods approach. However, authors only explain the qualitative part. No information can be seen about the qualitative part. From my point it could be relevant to include some analysis that allow seeing the relation between parents’ gender, studies or socioeconomic level and ideas about ASD.

R: A separate part on the quantitative and qualitative analysis were added and data on parents’ gender, studies or socioeconomic level and ideas about ASD added to the quantitative analysis part.

Finally, the discussion could benefit of the mentioned in the previous point about more studies regarding consequences of parents’ believes as well as the inclusion of qualitative methods. Otherwise limitations should be included about the lack of a quantitative approach  

R: Based on the quantitative analysis and data one paragraph was added to the discussion part and since this has been done no information in this regard mentioned in the limitations part.

Reviewer 2 Report

Some editing required may be that English is not their first language. The recruitment process needed to be clearer and identify the head of the centre contacted was to act as a gatekeeper ?? and interview process needs clarity, sounds as 2 interviews but only tell them for the 2nd what the study is - these impact on ethics of study. Also methods need to clarify the mixed methods nature of the study interviews were conducted and analysis using stats but this was on the same data ?? rather than a second phase.

Discussion I felt needed more on the culture aspect as this is the focus of the study. How do they compare or similar and what may be the point if similar as ASD has a communication/behavioural aspect which are most likely non-verbal and non-verbal language crosses all culture ??

Author Response

Second Reviewer:

Some editing required may be that English is not their first language.

R: The manuscript went through a proofreading process and corrections were made

The recruitment process needed to be clearer and identify the head of the centre contacted was to act as a gatekeeper ??

R: More information added to this part:

The head of these centres was contacted in person to explain the study and to get permission to participate in the weekly or biweekly parental sessions at the centre and to distribute announcements to recruit volunteer parents.

 and interview process needs clarity, sounds as 2 interviews but only tell them for the 2nd what the study is - these impact on ethics study.

R: In the first session parents were provided with the information sheet and data collecting questionnaires and in the second session the interview was done and the previously provided questionnaires were collected. Therefore, the first session was devoted to informing parents about the study its aims, answering their possible questions and giving them the demographic questionnaires about themselves and their children and helping the parents to get acquainted with the researcher and the study. The process and reasons for having two sessions were explained to prevent any possible ethical issues:

Also methods need to clarify the mixed methods nature of the study interviews were conducted and analysis using stats but this was on the same data ?? rather than a second phase.

R: More information provided regarding adopting a mixed method for the study and it was made clear that data out of interview considered for the qualitative analysis and data collected via demographic questionnaire was used for the quantitative analysis. Different sub-title was used for each part.

Discussion I felt needed more on the culture aspect as this is the focus of the study.

R: Another paragraph added on the cultural aspects

Acknowledgement or understanding of the parental cultural conception of developmental disability, as a sacred obligation, could have avoided misunderstandings around the aims of assessment and choice for intervention. Therefore, at very basic level findings of the present study might boost understanding of parental beliefs and ideas on ASD at a multicultural level and helping the different ASD services provider to predict justification and reasoning from a different lens.

How do they compare or similar and what may be the point if similar as ASD has a communication/behavioural aspect which are most likely non-verbal and non-verbal language crosses all culture ??

R: This point was added to the discussion part as a possible topic for further studies

Since ideas and impacts on ASD may differ in other socioeconomic classes future studies should target parents of children with ASD in other populations using bigger samples and other data collecting instruments that measure or address parental beliefs about autism to understand difference and similarities and what may be the point if similar as ASD has a communication and behavioural aspect which are most likely non-verbal crosses all cultures

Round 2

Reviewer 1 Report

Authors have adressed all my suggestions. 

I have no further comments.